# Recent Advances in the Delivery Carriers and Chemical Conjugation Strategies for Nucleic Acid Drugs

**DOI:** 10.3390/cancers13153881

**Published:** 2021-08-01

**Authors:** Shota Oyama, Tsuyoshi Yamamoto, Asako Yamayoshi

**Affiliations:** 1Chemistry of Functional Molecules, Graduate School of Biomedical Sciences, Nagasaki University, 1-14 Bunkyo-machi, Nagasaki-shi, Nagasaki 852-8521, Japan; bb55620002@ms.nagasaki-u.ac.jp (S.O.); tsuyoshi.yamamoto@nagasaki-u.ac.jp (T.Y.); 2PRESTO, Japan Science and Technology Agency (JST), 4-1-8 Honcho, Kawaguchi, Saitama 332-0012, Japan

**Keywords:** nucleic acid drugs, drug delivery system, conjugate, antibody

## Abstract

**Simple Summary:**

In recent years, nucleic acid drugs, such as antisense oligonucleotides (ASOs) and small interfering RNAs (siRNAs), have attracted attention as a new modality for cancer treatment. In this review, we introduce and discuss an overview of various drug delivery systems (DDSs) and ligand modification technologies that are being employed to improve the success and development of these drugs. It is our belief this review will increase the awareness of nucleic acid drugs worldwide and build momentum for the future development of new cancer-targeted versions of these drugs.

**Abstract:**

With the development of new anticancer medicines, novel modalities are being explored for cancer treatment. For many years, conventional modalities, such as small chemical drugs and antibody drugs, have worked by “inhibiting the function” of target proteins. In recent years, however, nucleic acid drugs, such as ASOs and siRNAs, have attracted attention as a new modality for cancer treatment because nucleic acid drugs can directly promote the “loss of function” of target genes. Recently, nucleic acid drugs for use in cancer therapy have been extensively developed and some of them have currently been under investigation in clinical trials. To develop novel nucleic acid drugs for cancer treatment, it is imperative that cancer researchers, including ourselves, cover and understand those latest findings. In this review, we introduce and provide an overview of various DDSs and ligand modification technologies that are being employed to improve the success and development of nucleic acid drugs, then we also discuss the future of nucleic acid drug developments for cancer therapy. It is our belief this review will increase the awareness of nucleic acid drugs worldwide and build momentum for the future development of new cancer-targeted versions of these drugs.

## 1. Introduction

For many years, the development of therapeutic drugs for cancer has been dominated by low molecular-weight chemical compounds. In this area cases exist in which drug discovery has been difficult, even when promising target molecules have been identified [1,2,3]. In recent years, however, nucleic acid drugs, such as antisense oligonucleotides (ASOs) and small interfering RNAs (siRNAs), have attracted attention as a new modality for cancer treatment [4,5,6]. These drugs can directly target genes and are potentially applicable to all types of diseases. Improvements in the technology used for artificial nucleic acid development have led to the successive approval of nucleic acid drugs for intractable and hereditary diseases; these drugs have been recognized worldwide for their therapeutic efficacy [4]. Nucleic acid drugs for cancer treatment are being actively developed, with many now at the clinical stage [7]. Thus, in the near future, it is expected that these drugs will contribute to the improvement of therapeutic outcomes as major cancer therapeutics.

Nucleic acid drugs are not readily permeable through cell membranes and often exhibit poor blood serum stability, rapid renal clearance and poor endosomal escape/cytoplasmic escape. Therefore, they are commonly used in combination with drug delivery system (DDS) carriers [8,9] (Figure 1). Initially, topically administered products for injection directly into the affected area were approved; however, subcutaneous and intravenous products are now being approved. ONPATTRO^®^ (patisiran), a siRNA drug with a liposomal formulation, was approved in 2018, exactly 20 years after the discovery of RNA [10]. Because nucleic acid drugs can be chemically synthesized like small molecule drugs, ligand-conjugated oligonucleotides have also attracted attention in recent years. Given the success of ligand-conjugated nucleic acids, an *N*-acetylgalactosamine (GalNAc)-conjugated siRNA drug (GIVLAAR^®^_,_ givosiran) has been developed by Alnylam (Cambridge, MA, USA), a leading company in the development of siRNA drugs [11]. This drug comprises tri-antennary GalNAc, a ligand of the asialoglycoprotein receptor that is highly expressed specifically in hepatic parenchymal cells, combined with siRNA; it can be transferred to hepatic parenchymal cells with high efficiency via subcutaneous administration and acts on a target gene [11,12,13]. Over the past years, development of siRNA drugs for cancer treatment have been conducted. To date, some of them, such as Atu027 [14,15,16] and siG12D-LODER [17,18,19], have passed or are currently in Phase II trials and they are expected to be eventually commercialized. A summary of each nucleic acid drug mentioned in this review is given in Table 1. In this review, we will focus on the delivery carriers of nucleic acid drugs for cancer therapy and provide an overview of DDS and ligand modification technologies that will contribute to the success and development of these drugs.

## 2. Nonviral Drug Delivery Systems for Nucleic Acid Drugs

The delivery of nucleic acid drugs can be divided into two main strategies: viral and nonviral delivery. Viral vectors are exceptionally efficacious in delivering genetic material to cells because millions of years of evolution have shaped and optimized them for this purpose. Recently, owing to developments in vector design and safety, viral gene therapy strategies have progressed toward clinical use against many genetic disorders. However, depending on the type of vector, viruses will always retain some of their inherent weaknesses, which can include potential immunogenicity, tumorigenicity, limited cargo-carrying capacity, and complex production. Importantly, viral vectors are not universally applicable to all nucleic acid-based molecules; for example, they are not compatible with the delivery of short synthetic oligonucleotides. Therefore, we focus on nonviral methods for the delivery of therapeutic oligonucleotides.

### 2.1. Cationic Vectors for Nucleic Acid Delivery

It is difficult for nucleic acids and their analogs to permeate cell membranes due to their negative-charged nature. Therefore, various positively charged molecules have been used as intracellular delivery carriers of therapeutic nucleic acids. Protamines are arginine-rich polycationic nuclear proteins that replace histones late in the haploid phase of spermatogenesis; they allow for denser packaging of DNA in the spermatozoon than would be possible with histones. This property has enabled protamines to be used as carriers for therapeutic oligonucleotides. Junguhans et al. were the first to demonstrate the cellular uptake of phosphodiester anti-c-*myc* antisense oligonucleotides into human promonocytic leukemia cells using protamines and to report their antisense effects [49]. Antisense oligonucleotide/protamine complexes have also been used successfully to inhibit human immunodeficiency virus 1 (HIV-1) gene expression [50].

Polyethyleneimine (PEI) was one of the first transfecting agents discovered; it is a cationic polymer that has been utilized as a polymeric agent for oligonucleotide administration [59,60]. PEI is an alkyl chain with primary, secondary, and tertiary amines. Only a portion of the amines in PEI is protonated at physiological pH; thus, it has a high buffering capacity and allows the release of nucleic acids in the acidic environment of the endosome via proton sponge effects [61]. Recently, a distinctive linear PEI derivative (jetPEI) has been shown to effectively facilitate intracellular DNA delivery; it is now in Phase 1/1b of clinical studies employing intratumoral/intralesional administration [51].

For cationic vectors, capillary embolization is a problem because red blood cell can become aggregated [62]. However, these aggregation problems can be overcome by covalent attachment of poly(ethylene glycol) (i.e., PEGylation) to the cationic vector. Merkel et al. reported that PEGylated-PEI containing partially chemically modified 25/27mer dicer substrate siRNAs (DsiRNAs) has systemic bioavailability after pulmonary application as well as an ability to knock down gene expression in the lungs [20]. On the other hand, introducing an anionic moiety to cationic vectors can effectively reduce hemolytic and cytotoxic effects. It has been reported that pentaerythritol-based anionic dendrimers can successfully deliver antisense oligonucleotides to cancer cells [21].

### 2.2. Liposomes and Lipid Nanoparticles 

Since the late 1980s, cationic liposomes have been considered as one of the most promising carriers for delivering nucleic acids to mammalian cells. To encapsulate nucleic acid drugs, such as siRNAs, in liposomes, it is necessary to create a core via the formation of cationic molecule and oligonucleotide complexes. Bickel et al. used PEI to form polyplexes with oligonucleotides and combined it with PEG-stabilized liposomes (PSLs) [22]. Upon intravenous administration, the DNA in PSLs was cleared from systemic circulation at a significantly slower rate than the rate at which naked PEI/oligonucleotide complexes were cleared. Furthermore, targeting of PSLs with antibodies specific to transferrin receptor has been shown to redirect biodistribution of the entrapped nucleic acid drugs, leading to significant accumulation in the targeted organ, i.e., the brain. Encapsulation of the PEI/oligodeoxynucleotide polyplexes within a long-circulating liposome provides a promising oligodeoxynucleotide delivery system for in vivo application.

Lipid nanoparticle (LNP) systems are currently the leading nonviral delivery systems for realizing the clinical potential of genetic drugs. Cullis et al. were the first to demonstrate the utility of LNPs based on ethanol injection to encapsulate antisense oligonucleotides [37]. In recent years, the world’s first siRNA drug and first nucleic acid drug with liposome implementation, namely patisiran, has been approved by the Food and Drug Administration (FDA) [10]. Patisiran consists of siRNA encapsulated in a LNP carrier (it was formerly known as a SNALP or “stable nucleic acid lipid particle”) [38,39]. The accumulation of SNALP within tissues of clinical interest takes advantage of passive disease-site targeting. 

The Phase 1 clinical trial of NBF-006, an LNP formulation with siRNA encapsulated, has been initiated [40]. Glutathione-S-transferase P (GSTP), which is overexpressed in various K-*ras* mutated cancers, such as lung and pancreatic cancers, has been selected as a target. Therefore, GSTP knockdown has been expected to be effective in treating those cancers. In addition, a Phase 2 clinical trial of BNT111, mRNA vaccine complexed with liposome, has been initiated in 2021 [41,42]. BNT111 contains four kinds of mRNAs and is expected to treat unresectable melanoma by inducing tumor-associated antigen-specific T-cell responses. With additional research, the development of LNPs equipped with nucleic acid drugs is likely to accelerate over time.

## 3. Conjugation of Functional Molecules to Therapeutic Oligonucleotides

In recent decades, the derivatization of nucleic acid drugs has been studied extensively. The nucleic acid cargo can be covalently attached to functional carrier molecules or loaded into supramolecular delivery devices. Conjugations of uptake-enhancing or targeting ligands to oligonucleotides provide the advantage of generating a defined molecule that allows for traditional pharmaceutical quality assessment. Several molecules have been attached to therapeutic oligonucleotides to improve their delivery, biodistribution, and cellular uptake; some are detailed in this section.

### 3.1. Cholesterol

Cholesterol was tethered to siRNA in one of the first reports of endogenous gene silencing in vivo; this was conducted under physiological conditions with a normal pressure injection in mice [63]. Cholesterol can easily be attached to a controlled-pore glass support prior to oligonucleotide synthesis, and an aminocaproic acid pyrrolidine phosphate linker is often used between ligands and oligonucleotides. Results have shown that cholesterol–siRNA conjugates can reduce the mRNA of targeted apoB by around 50% while unconjugated siRNA has no effect; similar results have been reported for the lipid docosanyl and stearoyl ligands [55]. Cholesterol–siRNA conjugates can also be used for noncovalent association to polymers, as demonstrated by in vivo gene silencing in combination with a targeted engineered polymer [56].

### 3.2. GalNAc

In 2019, Alnylam Pharmaceuticals, the company that developed patisiran, succeeded in developing an siRNA drug called “GalNAc-conjugated siRNA (GIVLAAR^®^, namely gibosiran)” [11,12]. This technology utilizes the binding of GalNAc to asialoglycoprotein receptors (ASGPR) that appear on the cell surface of hepatic parenchymal cells. Givosiran can be administered systemically (subcutaneously) without a carrier, whereas patisiran, which is encased in LNPs, requires a time-consuming intravenous infusion, making givosiran more useful in clinical practice. In addition, from the perspective of manufacturing and quality control, such conjugates are considered to be more advantageous than the drugs of this class with delivery carriers which often have complex structures like LNPs. In 2020, another GalNAc-siRNA (OXLUMO^®^, namely lumasiran) has been also approved by FDA [13].

GalNAc derivatives were first introduced to oligonucleotides by TsO’s research group in 1995 [23]. They developed GalNAc neoglycopeptide (ah-GalNAc)-conjugated oligodeoxynucleoside methylphosphonate (ah-GalNAc-oligo-MP) and successfully showed that the uptake of ah-GalNAc-oligo-MP by human hepatocellular carcinoma cells (Hep G2) is cell-type specific and can be completely inhibited by the addition of a 100-fold excess of free (ah-GalNAc)_3_ in the culture medium, indicating the cell uptake of ah-GalNAc-oligo-MP was ligand dependent. 

This specific and enhanced cellular uptake of GalNAc-conjugated oligonucleotides was also confirmed in vivo by several research groups [11,24]. Prakash et al. reported that antisense oligonucleotides conjugated to tri-antennary GalNAc improve the potency of therapeutic oligonucleotides about 10-fold in mice [24]. Now, there are various kinds of chemical modifications of GalNAc-conjugated, and from these reports, it has been shown that the GalNAc introduced into oligonucleotides does not necessarily have a tri-antennary structure, and, surprisingly, even mono-anntenary GalNAc-conjugation was also found effective [25,26]. In the future, we expect to uncover more detailed mechanisms of action of these monomeric GalNAc-conjugated oligonucleotides.

### 3.3. Folic Acid 

Folic acid (vitamin B9) binds with high affinity to the folate receptor protein to trigger cellular uptake via an endosomal pathway. The presence of the folate receptor on many cancer types has prompted the use of folate in targeted therapy [64]. Indeed, it has been used on liposomes or polyplexes to effectively deliver oligonucleotides to cancer cells that have the folate receptor [65,66]. Dohmen et al. were the first to develop folate-conjugated oligonucleotides, however, tethering folate to siRNA results in specific uptake but not silencing of reporter genes [67]. Folic acid–oligonucleotide conjugates are trapped in endosomes with insufficient endosomal escape to the cytosol for gene silencing. Later, Orellana’s group succeeded in eliciting the gene inhibitory effects of folic acid-conjugated oligonucleotides by connecting folic acid and oligonucleotides with a cleavable linker [57].

### 3.4. Cell Penetrating Peptides

Cell penetrating peptides (CPPs) can facilitate cellular uptake of their cargo, which is directly attached through covalent linkages or the formation of noncovalent complexes. When CPPs were first identified, they were derived from peptide sequences found in naturally occurring protein elements that exhibited inherent translocating properties. Some of these were important for subsequent CPP iterations including the transactivator of transcription from HIV [68], Penetratin-1 derived from the homeodomain of Antennapedia [69], transportan (a chimeric peptide derived from galanin and the wasp-venom peptide toxin mastoparan) [70], and cationic polyarginine and polylysine sequences such as Arg8 [71].

Within the context of CPP-mediated delivery, effector nucleic acids can either be directly conjugated to the CPP or noncovalently complexed, typically forming nanoparticle structures. Covalent conjugations of CPPs to charge-neutral oligonucleotides, such as peptide nucleic acids and phosphorodiamidate morpholino oligomer (PMO), have been examined extensively [58,72]. Indeed, PMOs are considered one of the most promising neutral-charge chemistries; they include a morpholine ring that replaces ribose and phosphorodiamidate linkages that replace phosphodiesters. Several methods can be used for conjugation of CPPs to PMOs, including maleimide linkage, disulfide linkage, click chemistry, and amide linkage; this process enhances the PMOs’ pharmacokinetic (PK) profile, biodistribution, and stability [58,72].

## 4. Antibody-Oligonucleotide Conjugates

Antibody–oligonucleotide conjugates (AOCs) belong to a class of chimeric molecules that combine within their structure two important biomolecules: monoclonal antibodies and oligonucleotides. Given the exceptional targeting capabilities of monoclonal antibodies and numerous functional modalities of oligonucleotides, AOCs have been successfully applied for a variety of purposes including imaging, detection, and targeted therapeutics. Here, we discuss the potential use of AOCs in cancer treatment.

### 4.1. Basic Composition and Functions of AOCs

Antibodies have the ability to recognize an antigen specifically and with high selectivity; thus, they can mark pathogens for further attack by various components of the immune system [73]. The exceptional selectivity of antigen recognition has resulted in their development into efficacious targeted therapeutics, both as single agents via antibody-dependent cell-mediated cytotoxicity (ADCC) and as vehicles for drug delivery, i.e., as antibody−drug conjugates (ADCs) [74,75].

AOCs are recognized as powerful tools for the therapeutic application of ADCs against various diseases [76]. In this system, the antibody is usually employed as a target recognition unit while the oligonucleotides play a variety of functional roles as therapeutic oligonucleotides, e.g., as siRNAs, aptamers, or antisense oligonucleotides. For therapeutic AOCs, the antibody can function as a delivery vehicle by increasing the circulation time of the oligonucleotide drugs in vivo [76].

Monoclonal antibodies have highly specific binding abilities to antigens via the Fab region [73]. In addition, the Fc region of antibodies plays a crucial role by expressing effector functions and increasing blood retention time. In general, lysine and cysteine residues are used for the antibody conjugation of functional molecules [77,78,79]. Since lysine residues are abundant on the surface of both the Fab and Fc regions, lysine-specific modification can disturb the antigen recognition of antibodies. Cysteine-specific modifications allow for site-specific introduction of functional molecules into antibodies because cysteine residues exist at the hinge region of antibodies [77,80]. However, the cleavage of disulfide bonds can potentially reduce the structural stability of antibodies and abrogate their function. Thus, use of these two methods must be chosen carefully depending on the intended application.

Several different methods for introducing functional molecules into antibodies have now been reported; these could potentially be applied in future reactions to introduce nucleic acid drugs into antibodies. Tagawa et al. reported the selective introduction of folic acid into the tryptophan residues of antibodies for induction of ADCC [81]. The tryptophan residue is the least abundant (around 1%) amino acid in the antibody and each residue has solvent accessibility because it is also the least surface-exposed proteinogenic amino acid. Antibody–folic acid conjugates were developed that showed significant cellular cytotoxicity toward folate receptor-expressing cancer cells via the ADCC mechanism. Another method involves the site-specific chemical conjugation of antibodies using an affinity peptide, IgG-BP, which can be intramolecularly cross-linked with a disulfide bond to the Fc site of the human IgG antibody [82]. This method enables rapid modification of a specific residue (Lys248 on Fc) in a one-step reaction under mild conditions.

### 4.2. Therapeutic Applications of AOCs

AOCs have several therapeutic applications. The clinical application of siRNA is often limited by the lack of efficient, cell-specific delivery systems. Song et al. were the first to report antibody-mediated siRNA delivery for the treatment of HIV/AIDS [43]. The fusion protein (F105-P) was designed with a protamine coding sequence linked to the heavy chain of a Fab fragment in an HIV-1 envelope antibody [43]. As mentioned in Section 2.1, protamines are small, arginine-rich, nuclear proteins that can form complexes with siRNA via electrostatic interaction. Song et al. demonstrated that siRNAs bound to F105-P induced silencing only in cells expressing the HIV-1 envelope. Following the publication of this study, research into the development of antibody–oligonucleotide conjugates has accelerated to the extent that many cases have now been reported.

Ma et al. investigated whether covalent or noncovalent constructs were more effective for siRNA delivery; covalent constructs have reductive disulfide linkers expected to undergo cleavage within endosomes whereas noncovalent constructs are based on the (D-arginine)9 (9r)-modified antibody [44]. Hu3S193, an anti-Lewis Y monoclonal antibody, was used for the development of the siRNA delivery vehicle in this study. Although both constructs were taken into the cells, the inhibitory effect of siRNA on gene expression was observed only in the noncovalent construct. It was speculated that the proton sponge effect of arginine residues may have been effective for the endosomal escape of siRNA.

Another example of a noncovalent construct is avidin–biotin technology, which has been applied for intracellular delivery of siRNA [45]. In this case, the siRNA was monobiotinylated to form a 1:1 construct with a streptavidin–monoclonal antibody conjugate (i.e., siRNA/SA/mA). An endocytosing monoclonal antibody to the transferrin receptor was used as the antibody for the siRNA/SA/mA construct, the intravenous administration of which caused a 69–81% decrease in luciferase gene expression in intracranial brain cancer in vivo. Thus, the delivery of siRNA to the brain following intravenous administration was made possible by receptor-specific antibody delivery systems and avidin–biotin technology.

Recently, studies have increasingly reported on covalent constructs of antibody–oligonucleotide conjugates. Glioblastoma stem cells (GSCs) are invasive and treatment-resistant brain cancer cells. Arnold et al. developed an antibody-conjugated, double-stranded, antisense oligonucleotide (dsAON) by click chemistry using an azide-modified antibody and an alkyne-modified dsAON [46]. They used antibodies against antigens expressed on the GSCs, such as CD44 and EphA2, and performed conjugation to chemically modified dsAONs. These therapeutic conjugates were able to successfully internalize, accumulate, and reduce target gene expression in GSCs. This report is the first to demonstrate the potential usage of antibody–oligonucleotide conjugates targeting cancer stem cells.

Sugo et al. reported an antibody–siRNA conjugate that targets cardiac and skeletal muscles [47]. Endothelial cells in the brain vasculature carry iron into the central nervous system via CD71-mediated transcytosis [48], which can be used to deliver drugs across the blood brain barrier. These authors developed anti-CD71 Fab′ fragment-conjugated siRNA, which produced significant gene-silencing effects in the gastrocnemius when injected intramuscularly. Interestingly, they examined several types of linkers for covalent conjugation of the anti-CD71 Fab′ fragment to siRNA and found that a non-cleavable linker (i.e., a maleimide linker) was effective whereas cleavable linkers (such as Val-Cit and DMSS linkers) did not improve silencing activity. These data suggest that low molecular-weight antibodies and fragments have considerable advantages when applied to endosomal release.

## 5. Exosome-Hijacking DDS

Exosomes are nano-sized extracellular vesicles that circulate in body fluids and act as a native transporting system for the delivery of cargo molecules from donor cells to recipient cells [83]. Exosomes naturally carry nucleic acids, such as DNA and RNA, to recipient cells, and thereby induce genetic modifications in both biological and pathogenic processes [84]. These features have brought exosomes into focus as potential endogenous carriers for the delivery of nucleic acid drugs to target cells [85]. Recently, we developed a novel strategy for capturing exosomes and delivering oligonucleotides to recipient cells, namely an “exosome-hijacking DDS” [27]. In this section, we provide an overview of exosomes and introduce our original DDS.

### 5.1. Properties of Exosomes

Exosomes are nano-sized (30–150 nm), lipid-bilayered, extracellular vesicles that can contain various molecules including proteins and lipids (Figure 2). The Exocarta database provides information on molecules that have already been identified in exosomes [86]. When they were first discovered, it was thought that exosomes transported waste in cells extracellularly; however, with the discovery of microRNAs in exosomes, they were redefined as carriers of materials between cells [84,87]. Despite around 40 years of research, not all of the functions and biological roles of exosomes are fully understood. Nevertheless, recent research on exosomes suggests that these naturally occurring carriers have the potential to deliver nucleic acids within our body. In future research, additional functions of exosomes will likely be revealed so that they can be used in DDS and disease treatments.

#### 5.1.1. Biogenesis and Cellular Uptake

The plasma membrane of cells inwardly invaginates to form endocytic vesicles known as early endosomes [88]. Additionally, the membrane of early endosomes invaginates to form more vesicles medially. These vesicles are known as intraluminal vesicles (ILVs), while the vesicles containing ILVs are known as multivesicular bodies (MVBs). When the membrane of the MVB fuses with the plasma membrane, the MVB exocytically releases its contents, which are commonly referred to as exosomes.

ILV formation and cargo sorting is regulated by endosomal sorting complex required for transport (ESCRT) proteins. Although the exact mechanism is not known, it has been reported that the ESCRT pathway interacts with ALIX, i.e., types of proteins involved in MVB formation, to sort tetraspanin proteins [89,90]. An overview of the ESCRT-dependent pathway has previously been summarized [91,92,93]; however, an ESCRT-independent pathway has also been reported [94,95]. It is possible that cells use different pathways to produce exosomes depending on the internal and external environment of cells or the cargo.

Exosomes secreted into body fluids (e.g., blood, urine, milk, and spinal fluid) or the supernatants of cultured cells are mainly taken up into cells by endocytosis, which is the main pathway of intracellular uptake and consists of several types of mechanism: clathrin-dependent endocytosis [96], caveolin-dependent endocytosis [97], macropino-cytosis [96,98], phagocytosis [99], and lipid raft-dependent endocytosis [100]. The intracellular uptake pathway of exosomes also differs depending on cell type and environment, similar to the exosome biogenesis pathway.

#### 5.1.2. Contents of Exosomes

Various molecules are contained within the vesicle and on the surface of the membrane. As detailed in a database of molecules identified in exosomes [86], tetraspanins (e.g., CD63, CD9, CD81, and CD82) [101,102,103,104], adhesion proteins (e.g., integrin and ICAM-1) [105], and HLA-G are found on the surface of exosomes [106]. On the other hand, heat shock proteins (e.g., HSP-70 and HSP-90) [107,108], MVB-forming proteins (Alix [109] and TSG101), microRNAs (miRNAs [110]), long noncoding RNAs [111], and circular RNAs [112,113] are found inside exosomes. Among these molecules, tetraspanins are used as exosome marker proteins since they are highly expressed on the surface membrane of exosomes secreted by many cell types. However, the expression pattern of tetraspanins and the size of exosomes differ depending on cell type. Zhang et al. investigated the heterogeneity of exosomes; they classified them into three subpopulations by size and investigated their properties [114]. Their data suggest that exosome heterogeneity is the most important issue for the practical application of exosome-based drugs.

### 5.2. Exosomes Used for the Delivery of Nucleic Acid Drugs

Recently, the number of approved nucleic acid drugs has increased, especially in the last five years. Chemical modifications can confer nucleic acids with stable structures and enhanced resistance to degradation by nucleases, but carriers are required for efficient and target-selective delivery of nucleic acids because they are easily degraded in the blood and rapidly eliminated from the kidneys. The challenge of developing efficient and organ-specific delivery methods for nucleic acids has been overcome by Alnylam Pharmaceuticals, which have produced GalNAc conjugates and received approval for three siRNA drugs [28,29].

As mentioned in Section 4.1, exosomes can incorporate nucleic acids and transport them to cells; the nucleic acids are contained inside the vesicle and stably transported despite the presence of nucleases in the blood. Therefore, exosomes can protect nucleic acid drugs from degradation and deliver them to target cells. In this section, we introduce some examples in which exosomes are used for the delivery of nucleic acids.

Erviti et al. encapsulated GAPDH siRNA in exosomes from self-derived dendritic cells [30]. Specifically, they fused a neuron-specific RVG peptide to the Lamp2b protein, which was expressed on the membrane of exosomes via engineering techniques. SiRNA was then electroporated into the exosomes before they were administrated intravenously. As a result, the expression of GAPDH was downregulated in several brain regions including the striatum, midbrain, and cortex. The authors estimated that the loading efficiency of siRNA into exosomes was about 20%, but additional research indicated that the true efficiency was <0.05% (the overestimation was likely caused by siRNA aggregation due to contamination of metal ions from the electrode used for electroporation) [31]. The results of this report facilitated the development of more efficient loading methods for nucleic acids.

Such efficient methods include hydrophobic modifications of siRNA, which improve the loading efficiency of siRNA to exosomes [32]. Didit et al. conducted a study in which siRNA targeting Huntingtin mRNA, which is the cause of Huntington’s disease, was loaded into exosomes by coincubation and incorporated into mouse primary cortical neurons. Hydrophobically modified siRNA (hsiRNA) consists of asymmetric oligonucleotides and contains cholesterol at the 3′ end of the passenger strand for improved stability and cellular internalization. This cholesterol modification enabled the efficient loading of siRNA and its uptake by cells. Additionally, exosome-associated hsiRNA caused a 75% reduction in Huntingtin mRNA in a dose-dependent manner. This strategy may be applicable to other types of nucleic acids.

Zhao et al. developed biomimetic nanoparticles, which they named “CBSA/siS100A4@Exosome,” and they successfully downregulated the cellar growth of metastatic triple-negative breast cancer [33]. CBSA/siS100A4@Exosome comprises cationic bovine serum albumin (CBSA), siRNA targeting S100A4 (siS100A4, which relates to tumor metastasis and progression), and exosomes recovered from the supernatants of 4T1 breast cancer cells. SiS100A4 complexed with CBSA was successfully incorporated into the exosome with a siRNA loading efficiency of 86.7%. When CBSA/siS100A4@Exosome treatment was applied, S100A4 expression levels were downregulated in vitro and the number of metastatic nodules in the lungs was greatly reduced in vivo.

Munagala et al. developed a novel DDS based on exosomes using a surface modification method [34]. Exosomes from bovine milk were functionalized with folic acid and modified with PEI. The generated complexes, which were named EPMs, interacted with nucleic acids on the surface of exosomes and formed a ternary complex. The authors assessed the silencing efficiency of EPM equipped with siRNA targeting K-*ras* (siKRAS) and the sensitivity of paclitaxel when p53 plasmid DNA was delivered into p53-knockout mice. Consequently, expression levels of KRAS were reduced by about 50%–80% and lung tumor growth was also downregulated by about 70%. Furthermore, p53 was expressed in p53-knockdown mice and the sensitivity of paclitaxel was also recovered. On the other hand, in 2021, a Phase 1 clinical trial of siRNA encapsulated in mesenchymal stem cell (MSC)-derived exosomes has been started in patients with metastatic pancreatic ductal adenocarcinoma (PDAC) with K-*ras* G12D mutation [35,36]. Overall, the findings of several studies, including those discussed above, show that exosomes are capable of delivering nucleic acid drugs located inside or outside of exosomes.

### 5.3. Antibody-Oligonucleotide Conjugates Targeting microRNAs in Exosomes: ExomiR-Tracker

MiRNAs are small noncoding RNAs that bind to mRNAs and thereby regulate their expression. Lim et al. showed that miRNAs can regulate many target mRNAs [115]. According to miRbase, a database of miRNAs, >2500 miRNAs have been identified in humans [116]. These miRNAs are thought to regulate >60% of all human genes and to play key roles in gene expression and cell proliferation [117]. Exosomes have an abundance of miRNAs; thus, they contribute to regulating gene expression in exosome-recipient cells. Notably, miRNAs also exist in exosomes secreted from cancer cells and contribute to the formation of the premetastatic niche and cancer cell migration [118,119]. The organotroph of exosomes is determined by the expression pattern of integrins on their surface [120] and this feature of exosomes contributes to their drug delivery capabilities. Several cancer type-specific miRNAs have been identified; these have become therapeutic and diagnostic targets for cancer treatment [121,122].

To inhibit the function of exosomal-miRNA, antisense oligonucleotide complementary to exosomal-miRNA (i.e., anti-miRNA) is commonly used. Our group have focused on the surface molecules of exosomes and we have attempted to develop a method for delivering anti-miRNA to exosome-recipient cells using anti-exosome antibody-anti-miR complexes (a system known as “ExomiR-Tracker”) [27] (Figure 3). First, we assessed the intracellular uptake of Alexa647-labeled anti-exosome antibodies using a confocal laser scanning microscope. As antigens on the exosome membrane, CD63, CD9, and CD81 (all exosome marker proteins) were selected. Anti-CD63 antibody was incorporated to a large extent into Cal27 (oral squamous carcinoma) cells; thus, we synthesized ExomiR-Tracker using anti-CD63 antibody. The antibody was thiolated with Traut’s reagents and oligo-arginine peptides were continuously introduced to its thiol groups. The arginized antibody was complexed with anti-miRNA in phosphate-buffered saline to produce ExomiR-Tracker. In addition, we used anti-miRNA containing 22 nucleotides, of which seven nucleotides were replaced with locked nucleic acids (2′,4′-bridged nucleic acids), to stabilize the system in vivo by improving its resistance to nucleases [123]. Miravirsen, an anti-miRNA drug containing locked nucleic acids, has shown strong results in a Phase 2 study [124].

Furthermore, we assessed the intracellular uptake of ExomiR-Tracker equipped with fluorescence-labeled anti-miRNA via confocal laser scanning microscopy. Fluorescence-labeled anti-miRNA was found to have been successfully incorporated into Cal27 cells by ExomiR-Tracker. In addition, the functional inhibition of miRNA-21 by ExomiR-Tracker equipped with anti-miRNA-21 was evaluated with a luciferase reporter assay, the results of which showed that ExomiR-Tracker was able to inhibit the function of target miRNA-21 in a sequence-specific manner. Finally, we subcutaneously coinjected Cal27 cells and ExomiR-Tracker into the hind foot of nude mice and assessed the antitumorigenic effects in vivo. Surprisingly, the tumor volume of mice treated with ExomiR-Tracker was small compared with the tumor volume of untreated mice. Thus, ExomiR-Tracker seems to functionally inhibit tumorigenesis in vivo.

## 6. Application to Cancer Treatments Using Surface Molecules on Exosome Membranes

Our ExomiR-Tracker strategy suggests the possibility that various technologies other than drug discovery systems could be developed using the surface molecules on exosomes without impairing the functions of exosomes. In this section, we discuss several studies in which the surface molecules of exosomes, such as tetraspanins and membrane lipids, were used.

### 6.1. Phosphatidylserine

Phosphatidylserine is an important type of phospholipid that constitutes the lipid bilayer of the plasma membrane. Some phospholipids, such as phosphatidylserine, phosphatidylcholine, phosphatidylethanolamine, and sphingomyelin, consist of lipid bilayers, but the composition of membrane lipids differs greatly between the inner and outer cellular membrane. The inner membrane contains phosphatidylserine and phosphatidylethanolamine, whereas the outer membrane contains phosphatidylcholine and sphingomyelin. This asymmetric composition of lipids is regulated by membrane proteins such as flippase and the scramblase [125]. However, it has been shown that phosphatidylserine in apoptotic cells is exposed to the outer membrane where it is recognized as an “eat me” signal by macrophages and phagocytosed [126,127]. In addition, the phosphatidylserine of exosomes is reportedly in the outer membrane.

Kooijman et al. focused on the expression of phosphatidylserine on the surface of exosomes and epidermal growth factor receptor (EGFR) on the plasma membrane of cancer cells; thereby, they developed a tumor-targeting strategy known as the “plug-and-play approach” [128]. Specifically, they generated a fusion protein (EGa1–C1C2) consisting of the phosphatidylserine-binding domain (i.e., C1C2) of lactadherin and an anti-EGFR nanobody (i.e., EGa1; a nanobody is a new type of antibody drug with a high affinity to the antigen that can be generated using *Escherichia coli* since it constitutes the variable domain of the antibody’s heavy chain). They showed that EGa1–C1C2 selectively bound to phosphatidylserine among membrane lipids such as phosphatidylcholine, phosphatidylethanolamine, and sphingomyelin. Furthermore, EGa1–C1C2 enabled the incorporation of extracellular vesicles into A431 cells, which have EGFR on their plasma membrane, even when Neuro-2A cells, which do not have EGFR, were present in excess. The plug-and-play approach can provide exosomes with tumor-targeting abilities and facilitate the uptake of exosomes with reporter proteins or drugs.

### 6.2. Tetraspanin Proteins

CD63, CD9, and CD81 are proteins that are highly expressed on the surface membrane of exosomes. As described above, these proteins belong to the tetraspanin protein family, 33 species of which have been identified in humans. CD63 protein, which we utilized for ExomiR-Tracker, forms a complex with other proteins and functional molecules on the membrane of exosomes; thereby, it constructs a localized functional microdomain known as the tetraspanin-enriched microdomain. CD63 contributes to regulating intercellular adhesion and fusion through the tetraspanin-enriched microdomain region [104,129,130]. It has been shown that CD63 is localized 7-fold more in ILVs than in late-endosomes [131]. Along with CD63, CD9 forms the tetraspanin-enriched microdomain to affect signal transduction among cells and cell adhesion [132,133,134]. Moreover, CD9 has many known relationships with cancer cells [135]. For example, Lu et al. reported that the tumorigenicity of pancreatic cancer was reduced via the inhibition of alpha-secretase activity by anti-CD9 antibody or CD9 knockdown [136]. CD9 is also known to contribute to the immune system [137]. Thus, CD9 affects many processes in the body making it an attractive molecule for further investigation and development as a therapeutic target [138].

Yoshioka et al. developed a high-speed and high-sensitive tool named “ExoScreen” to detect exosomes without purifying the blood of colorectal cancer patients [139]. In this method, two antibodies that respectively recognize CD9 and CD147 are used. Only when the two antibodies are close together can exosomes in the blood be directly detected by the release of a fluorescent signal. The authors found that the number of exosomes coexpressing CD9 and CD147 was significantly higher in colorectal cancer patients than that in healthy subjects. By changing antibodies to disease-specific antigens on exosome membranes, ExoScreen can be applied to detect other cancer types. As a leading diagnostic tool for cancer, further development of ExoScreen is expected.

## 7. Conclusions

With the development of anticancer medicines, new cancer treatment modalities are being explored. Conventional forms, such as small chemical drugs and antibody drugs, work by “inhibiting the function” of target proteins. In this review, we have introduced nucleic acid drugs, such as ASOs and siRNAs, which promote the “disappearance” or “loss of function” of the target protein and act by new mechanisms that utilize the inherent characteristics of oligonucleotides. Although there are no nucleic acid drugs approved for cancer treatment yet, recent results of several clinical trials suggest that anti-cancer nucleic acid drugs will probably be approved in the near future. Furthermore, it is expected that nucleic acid drugs will be developed and practically used in a coordinated manner according to the characteristics of cancer types. It is our hope that this review will increase the awareness of nucleic acid drugs worldwide and build momentum for the future development of new cancer-targeted versions of these drugs.

## Figures and Tables

**Figure 1 cancers-13-03881-f001:**
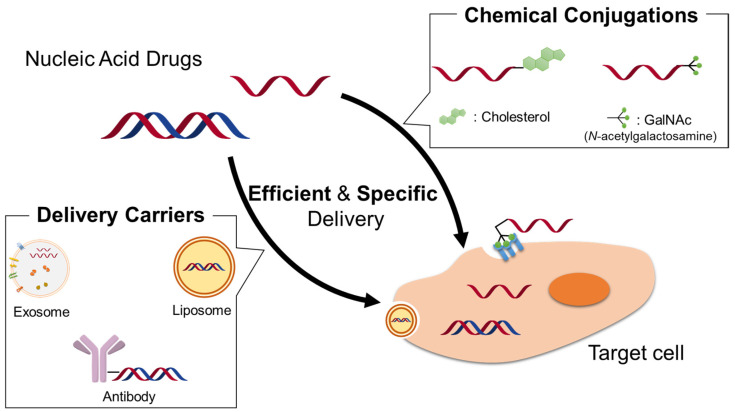
Schematic illustrations of delivery carriers and chemical conjugation strategies for nucleic acid drugs. Here shows typical delivery strategies: cholesterol conjugations [20,21,22], GalNAc conjugations [11,12,13,23,24,25,26], exosome [27,28,29,30,31,32,33,34,35,36], liposome [14,15,16,22,37,38,39,40,41,42], antibody [27,43,44,45,46,47,48].

**Figure 2 cancers-13-03881-f002:**
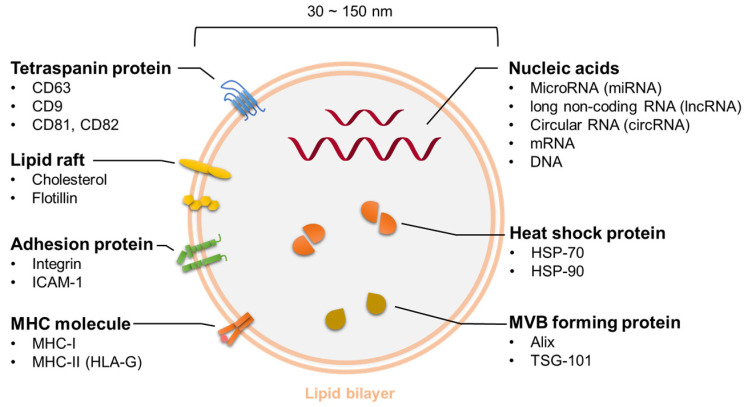
Various molecules are contained within the vesicle and on the surface of the membrane.

**Figure 3 cancers-13-03881-f003:**
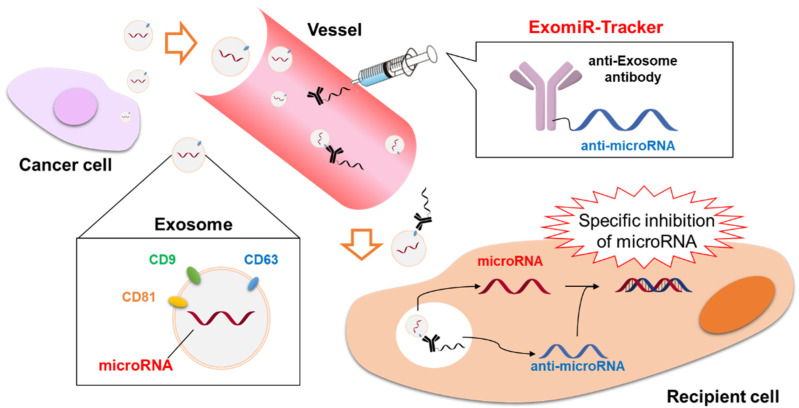
Schematic representation of the concept of “ExomiR-Tracker” [43].

**Table 1 cancers-13-03881-t001:** Drug delivery systems (DDS) for nucleic acid drugs, including those in clinical trials.

Carrier	Molecule	Target	Disease	Approved	Clinical Trial		References
					Phase (Approved)	ClinicalTrials.gov Identifier	
Protamine Protamine	ASO ASO	c-*myc* HIV-1	Histiocytic lymphoma cell HIV-AIDS	- -	- -	- -	[49] [50]
PEI PEI	DNA vaccine RGT100	- RIG-1	B-cell non-Hodgkin’s lymphoma Advanced metastatic solid tumor	- -	I/ongoing I/completed	ISRCTN31090206 NCT03739138	[51] [52]
Anionic dendrimer	ASO	EGFR	Epidermoid carcinoma	-	-	-	[21]
LNP Lipoplex Lipoplex Liposome PSL	siRNA siRNA mRNA vaccine siRNA DNA	TTR PKN3 - GSTP -	hATTR amyloidosis advanced pancreatic cancer (cancer vaccine) Non-small cell lung cancer (brain targeting)	Yes, patisiran - - - -	(2018) II/completed II/recruiting I/recruiting -	- NCT01808638 NCT04526899 NCT03819387 -	[10,38,39] [14,15,16] [41,42] [40] [22]
EDV	miR-16 mimic	EGFR	Malignant pleural mesothelioma Non-small cell lung cancer	-	I/completed	NCT02369198	[53,54]
LODER™	siRNA	K-ras G12D	Pancreatic cancer	-	II/recruitig	NCT01676259	[17,18,19]
GalNAc GalNAc Cholesterol Folic acid (FA) CPP	siRNA siRNA siRNA microRNA PMO	ALAS1 HAO1 ApoB - c-myc	Acute hepatic porphyria Primary hyperoxaluria type 1 (liver targeting) Breast cancer (enhance PMO’s PK)	Yes, givosiran Yes, lumasiran - - -	(2019) (2020) - - -	- - - - -	[11] [13] [55,56] [57] [58]
Anti-body Antibody Fab’ + protamine	siRNA dsASO siRNA	STAT3 DRR/FAM107A c-myc, VEGF	Lewis-y positive cancer cell Glioblastoma stem cell HIV/AIDS	- - -	- - -	- - -	[44] [46] [43]
Antibody + exosome	ASO	miR-21	Adenosquamous carcinoma	-	-	-	[27]
Exosome Exosome Exosome MSC-derived exosome Exosome + FA + PEI	siRNA hsiRNA siRNA siRNA siRNA/plasmid	GAPDH Huntintin S100A4 K-ras G12D K-ras/p53	Alzheimer’s disease Huntington’s disease TNBC Pancreatic cancer Lung cancer	- - - - -	- - - I/recruiting -	- - - NCT03608631 -	[30] [32] [33] [35,36] [34]

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
