# Peer review of "Recent Advances in the Delivery Carriers and Chemical Conjugation Strategies for Nucleic Acid Drugs"

_cancers, 2021, doi:10.3390/cancers13153881_

Round 1

Reviewer 1 Report

no further comments

Reviewer 2 Report

The work was now improved and better undestandable. Thus, it is worth to be published. 

This manuscript is a resubmission of an earlier submission. The following is a list of the peer review reports and author responses from that submission.

Round 1

Reviewer 1 Report

The revision work is interesting and quite exhaustive. However I have two important remarks:

1) It lacks of a table, with all the carriers - molecules - target - application, resuming the work you have described in the text.

2)The conclusions are not appropriate, especially as regards the sentence "Although many nucleic acid drugs for cancer have already been developed, practi cally none have been approved for use in cancer therapy". If you see the work: "Lipid Delivery Systems for Nucleic-Acid-Based-Drugs: From Production to Clinical Applications", there is a list (Table 1) about the siRNA-lipid delivery systems used in clinical trials, updated to 2019.  Thus, you must search for evolutions about these clinical trials, or other ones. 

Moreover,  It is necessary  a table with all the abbreviations in the text.

Author Response

Answer to Reviewer 1

The authors are grateful to you for your careful reading and comments on our manuscript (cancers-1214117). Here are our answers to your comments.

(1) It lacks of a table, with all the carriers - molecules - target - application, resuming the work you have described in the text.

We agree with your comment and deeply thank for your accurate suggestion. We added the Table 1 showing the combinations of carriers, molecules, targets, and applications that we have described in the text (Table 1. Drug delivery systems (DDSs) for nucleic acid drugs, including those in clinical trials.).

(2) The conclusions are not appropriate, especially as regards the sentence "Although many nucleic acid drugs for cancer have already been developed, practically none have been approved for use in cancer therapy". If you see the work: "Lipid Delivery Systems for Nucleic-Acid-Based-Drugs: From Production to Clinical Applications", there is a list (Table 1) about the siRNA-lipid delivery systems used in clinical trials, updated to 2019. Thus, you must search for evolutions about these clinical trials, or other ones.

We are sorry for unsuitable statement for the conclusion and completely agree with your comment. We obtained information from the work (“Lipid Delivery Systems for Nucleic-Acid-Based-Drugs: From Production to Clinical Applications”) about siRNA-lipid delivery systems used in clinical trials. Furthermore, we surveyed more information on clinical trials for nucleic acid drugs and added a summary of them in Table 1 as mentioned above (1).

We changed unsuitable conclusions in the abstract section and conclusion section and highlight the changes in the revised manuscript by using colored text.

(3) It is necessary a table with all the abbreviations in the text.

Thank you so much for your precise pointing out. We added the list with all the abbreviations at the end of revised manuscript.

Reviewer 2 Report

While there has been an explosion of research into gene therapy for the treatment of neurodegenerative diseases, the translation of this technology for cancer therapy is lagging. The authors state that the aim of this review is to increase awareness of gene therapy in cancer the community, however many recent reviews on this topic already exist. For example see PMID: 30574185, PMID: 30915230, PMID: 28094775, PMID: 27582234, PMID: 31100411. It is clear that there are bottlenecks in the clinical translation of nucleic acid-based therapies, unfortunately, the authors fail to address the primary reasons for this, such as scale up of manufacture, cost and toxicity, or present any possible solutions to overcome these challenges in order to drive the field forward.

Furthermore the review focuses on  antisense oligonucleotides and siRNA, however, additional nucleic acids such as plasmids, miRNA, DNA, gRNA  have been used for cancer therapy but are not comprehensively  reviewed here– by omitting this information the authors limit the scope, and the potential impact, of their review.

In the introduction, the authors justify the need to develop innovative nanocarriers in order to effectively deliver RNAi to cells due to their limited ability of nucleic acid drugs to cross the cell membrane. However a number of additional important chemical and physical barriers exist that also effect delivery of nucleic acids and should be considered when designing carriers and have not been mentioned here, including compromised blood serum stability, rapid renal clearance, off-target effects low transfection efficiency, poor endosomal escape/cytoplasmic escape.

The review needs to directly address these major concerns as well as the specific comments below before is it suitable for publication.

  1. Describe GOF mutations, and specifically in the context of cancer, the role oncogenes and tumour suppressor genes play in the initiation and progression of cancer. By doing this, the role of RNAi as a therapeutic strategy becomes justifiable.
  2. the section on cationic polymers and liposomes/nanoparticles appears to not be comprehensive or focused solely on examples outside the cancer field i.e. drug delivery systems that pass the BBB to treat neurodegenerative diseases such as Huntington’s or Alzheimer’s disease. While the lessons learnt in other fields can be used to inform the cancer field, not all aspects are relevant. The authors should carefully consider how advances in other fields can be specifically translated to the cancer field.
  3. Figure 1, in the chemical modification box on right hand side, what are these modifications? Specifically list/ label them here or in the figure legend.
  4. Figure 1, in the delivery carriers box add Ab and nanocarrier labels to this figure- also include exosmes in this box as an additional nanocarrier.
  5. Introduce the full term before using the abbreviation throughout i.e. siRNA and ASO.

Author Response

Answer to Reviewer 2

The authors are grateful to you for your careful reading and comments on our manuscript (cancers-1214117). Here are our answers to your comments.

(1) Describe GOF mutations, and specifically in the context of cancer, the role oncogenes and tumor suppressor genes play in the initiation and progression of cancer. By doing this, the role of RNAi as a therapeutic strategy becomes justifiable.

Thank you for your accurate suggestions and we deeply agree with your comments. As you mentioned, some nucleic acid drugs for cancer therapy have currently been under investigation in clinical trials; we therefore added description of GOF mutations in the context of cancer. Furthermore, in this review, we focused on providing an overview of carriers to deliver nucleic acid drugs to target organs or cells, not only in the field of oncology, but in all aspects of disease treatment. Since there are still very few examples of FDA-approved nucleic acid drugs, we think it is important to summarize the latest DDS for nucleic acid drugs regardless of cancer treatment. Therefore, we have mainly described about DDSs that have been reported so far, and clinical trials of nucleic acid drugs for cancer treatment are also described in the main text. We thank again for your precise advice in this matter.

(2) The section on cationic polymers and liposomes/nanoparticles appears to not be comprehensive or focused solely on examples outside the cancer field i.e. drug delivery systems that pass the BBB to treat neurodegenerative diseases such as Huntington’s or Alzheimer’s disease. While the lessons learnt in other fields can be used to inform the cancer field, not all aspects are relevant. The authors should carefully consider how advances in other fields can be specifically translated to the cancer field.

We completely agree with your comment and deeply thank again for your pointing it out precisely. We took your point and thought we should carefully have considered whether the lessons learnt in other fields can similarly be used in cancer field. For more meaningful discussing, we introduced information about clinical trials for anti-cancer nucleic acid drugs, such as siRNA-encapsulated LNP in the section 2.2 and siRNA-encapsulated exosome in the section 5.2, and highlight by using colored text.

(3) Figure 1, in the chemical modification box on right hand side, what are these modifications? Specifically, list/ label them here or in the figure legend.

Thank you so much for your comment and we apologize for the inappropriate manuscript preparation. In the chemical modification box in Figure 1, left side motif (combined hexagons and pentagon) and right side motif (trident) represent cholesterol and N-acetylgalactosamine (GalNAc), respectively. Furthermore, we added the label in the box and in the figure legend with each references.

(4) Figure 1, in the delivery carriers box add Ab and nanocarrier labels to this figure- also include exosomes in this box as an additional nanocarrier.

We agree with your comment and deeply thank again for your accurate suggestions. In the delivery carriers box in Figure 1, we included exosomes in this box and added the labels of them. Moreover, we added the figure legend of Figure 1 and highlight by using colored text.

(5) Introduce the full term before using the abbreviation throughout i.e. siRNA and ASO.

We are sorry again for the careless preparation of our manuscript. We introduced the full term before all the abbreviations in the text. Furthermore, we added the list about all the abbreviations and highlight by using colored text in the end of revised manuscript.

(6) Furthermore the review focuses on antisense oligonucleotides and siRNA, however, additional nucleic acids such as plasmids, miRNA, DNA, gRNA have been used for cancer therapy but are not comprehensively reviewed here– by omitting this information the authors limit the scope, and the potential impact, of their review.

We agree with your comment and deeply thank again for your precise pointing out. As you mentioned, plasmids, miRNAs, and DNAs have also been developing for cancer treatment now. Although we would like to summarize all of them, this review focuses on nucleic acid drugs that have already been approved or undergoing clinical trials, rather than basic research. As a result, we have decided to focus on antisense nucleic acids and siRNA. Furthermore, DNAs and gRNAs (used in CRISPR/Cas9 system) are mainly categorized as “gene therapy”, not as “nucleic acid drugs”, therefore we did not emphasize DNAs and gRNAs in this review. DDSs are essentially required not only for nucleic acid drugs, but also for other therapeutic nucleic acids generally used in gene therapy. In near future, we believe that clinical trials of therapeutic nucleic acids with DDS for use in the field of gene therapy would be accumulated.

(7) In the introduction, the authors justify the need to develop innovative nanocarriers in order to effectively deliver RNAi to cells due to their limited ability of nucleic acid drugs to cross the cell membrane. However a number of additional important chemical and physical barriers exist that also effect delivery of nucleic acids and should be considered when designing carriers and have not been mentioned here, including compromised blood serum stability, rapid renal clearance, off-target effects low transfection efficiency, poor endosomal escape/cytoplasmic escape.

We completely agree with your comment and deeply thank again for your precise suggestions. As you mentioned, in the practical application of nucleic acid drugs and their carriers, there are issues to be solved such as clearance and poor endosomal escape as well as cell membrane permeability. We added the statement about above in the introduction section.

Recently, chemically modified nucleic acid, such as oligo(nucleoside phosphorothioate)s (S-Oligos), morpholino nucleic acids, 2’-methoxyethyl-modified oligonucleotides have been utilized to improve the blood serum stability and off-target effects. We would like to summarize all the points you've suggested, however, in this special issue "Exosome Biology for Nucleic Acid Medicine—From Bench to Bed", we plan to publish three reviews (I’m also a guest editor of this special issue). In one of the three reviews, a researcher from a pharmaceutical company will summarize in detail about the chemical modifications of nucleic acid drugs that are being applied clinically. He also discussed such issues to be solved such as clearance and poor endosomal escape as well as cell membrane permeability. In this review, we have focused on the carriers for nucleic acid drugs and summarized the basic findings of those typical carriers. We thank again for your precise advice in this matter.

Reviewer 3 Report

The authors did a good job in writing a comprehensive paper regarding an increasing field of pharmacology.

Author Response

Thank you so much for your time to review our manuscript.